# Conformal Prediction with Model-Aware Debiasing

## Abstract

Bias in model estimation can lead to wider prediction intervals, diminishing the utility of predictive inference. Existing methods have attempted to address this issue, but they often rely on nontrivial assumptions such as specific error distributions or model sparsity, and fail to guarantee coverage in finite samples, which makes their predictions unreliable in practice. To overcome these limitations, we propose a model-aware conformal prediction method that utilizes known model information to achieve debiasing while leaving the unknown aspects, such as data distribution, to the conformal prediction framework. This approach requires only the assumption of exchangeability, making it broadly applicable across various models. Importantly, it retains the finite-sample coverage property and produces shorter prediction intervals compared to existing methods. When applied to threshold ridge regression, we theoretically demonstrate that the model-aware conformal prediction maintains finite-sample marginal coverage and, under certain assumptions, converges to the oracle prediction band, achieving asymptotic conditional validity. Numerical experiments further show that our method outperforms existing methods, providing more efficient prediction intervals across diverse regression datasets.

## 1 Introduction

Uncertainty quantification is crucial in developing machine learning models, particularly in contexts involving high-stakes decision-making in fields such as medicine and finance. However, a key challenge in constructing effective prediction bands is the bias in model estimation. Whether introduced by model assumptions, data sparsity, or other factors, bias often leads to overly conservative prediction bands that unnecessarily widen to account for the model's systematic errors. This results in less informative prediction bands and reduces their utility in practice. Many existing methods seek to address this issue by constructing prediction bands that account for model bias(Zhang & Zhang (2014); Javanmard & Montanari (2014); Van de Geer et al. (2014); Zhang & Politis (2022; 2023)). However, these methods all rely on nontrivial assumptions such as the error distribution, the homoscedasticity of errors, the quality of the estimator, sparsity, and low intrinsic dimensionality, which are often not true in practice. In addition, most of them obtain asymptotic but not finite-sample validity.

To overcome these limitations, we propose the model-aware conformal prediction, a novel approach integrating the underlying mechanisms of the model and parameters to alleviate the influence of bias on prediction inference, and leaving the unknown aspects—such as the distribution of the data—to the conformal prediction framework to maintain the finite-sample coverage property. Specifically, We account for the bias when constructing the nonconformity score function. Then we take the threshold ridge regression as an example to illustrate its better performance, since it is computationally simple and may be preferable in some settings(Zhang & Politis (2022) and Shao & Deng (2012)).

In summary, our contribution are as follows:

- We propose a model-aware conformal prediction framework, which provides shorter and more efficient prediction intervals while retaining the finite-sample coverage property across a wide range of applications.

- We apply our method to threshold ridge regression and theoretically demonstrate that, under certain conditions, it converges to the oracle prediction band, and achieves asymptotical conditional validity.

- Through experiments on real-world datasets, we show that model-aware conformal prediction produces shorter prediction intervals while maintaining the required coverage than existing methods.

## 1.1 RELATED WORKS

In recent years, the inference problem in high-dimensional models has garnered significant attention, though much of the focus has been on regression coefficients. For instance, Zhang & Zhang (2014) assumed that the linear model is correct and constructed confidence intervals for individual coefficients $\beta_j$ using debiased estimators obtained by inverting the KKT conditions of $\ell_1$-penalized regression problems. Similar methods are discussed in Javanmard & Montanari (2014) and Van de Geer et al. (2014). Zhang & Politis (2022) proposed the debiasing method for the estimator in threshold ridge regression and used Bootstrap to construct prediction intervals. Zhang & Politis (2023) extended it to linear models with heteroskedastic and correlated errors. Another prominent approach in high-dimensional inference is post-selection inference(Liu & Yu (2013); Berk et al. (2013); Lee et al. (2016); Tibshirani et al. (2018); Zrnic & Jordan (2023)), which first applies variable selection techniques to identify influential covariates, followed by fitting ordinary least-squares regression on the selected covariates. These methods focus on providing coverage for coefficients in the best linear approximation given the selected covariates. However, these approaches often rely on nontrivial assumptions, such as specific error distributions, homoscedasticity of errors, the quality of the estimator, sparsity, and low intrinsic dimensionality, which are frequently violated in practice. When these assumptions do not hold, the inference tools become invalid, especially in cases of model misspecification.

In contrast, conformal inference does not depend on such stringent conditions, making it particularly well-suited for high-dimensional settings where traditional assumptions are often unrealistic. There have been several attempts to conformalize the high-dimensional models. Hebiri (2010) proposed a partial conformalization of LASSO; however, this approach did not provide coverage guarantees. Lei (2019) introduced a piecewise linear homotopy method for LASSO to construct prediction bands efficiently and extended this technique to the elastic net framework. Izbicki et al. (2022) considered the conditional density function as a nonconformity score function and utilized a data-driven partition that scales to high dimensions. However, these methods do not adequately address the impact of estimation bias in the models, which can significantly affect their efficiency.

Our work is related to a recent work by Zhang & Politis (2022), which used a bias correction method in threshold ridge regression to improve the performance of prediction inference. They proposed the hybrid bootstrap to construct prediction intervals and established the consistency property of the prediction region, but in an asymptotic sense, which is often insufficient in practice. This paper uses the same debiasing technique as Zhang & Politis (2022) but extends it into the conformal prediction framework, offering several advantages. First, our approach ensures finite-sample marginal coverage without depending on strong assumptions, making it more robust and practical in real-world applications. Second, we theoretically prove that the prediction intervals constructed by our method converge to the oracle prediction band in Lei et al. (2018) under certain conditions and have asymptotic conditional validity.

## 1.2 ORGANIZATION

The paper is organized as follows. Section 2 includes notations, model setup, assumptions, and a brief review of conformal prediction. In Section 3, we propose conformal prediction with model-aware debiasing, apply it to the threshold ridge regression framework, and present the corresponding theoretical results. We demonstrate its performance with numerical experiments, comparing with the standard conformal prediction and the bootstrap method in Section 4. Section 5 contains further remarks and future directions.

## 2 PRELIMINARIES

### 2.1 NOTATION

Consider the following regression model:

$$\mathbf{y} = \mu(\mathbf{X}) + \boldsymbol{\epsilon} \tag{1}$$

The $(n + 1) \times p$ design matrix $\mathbf{X}$ is assumed to have rank $r$. $\mathbf{X}$ includes $n + 1$ pairs, $(X_1, Y_1), \ldots, (X_n, Y_n)$ and $(X, y)$ where $(X_1, Y_1), \ldots, (X_n, Y_n)$ are observations and $X$ is a given data, $y$ is unknown. Sometimes we write $(X, y)$ as $(X_{n+1}, y)$ for convenience.

The error vector $\boldsymbol{\epsilon}$ has mean zero and satisfies assumptions to be specified later.

To analyze the efficiency of the prediction band, we first collect some common assumptions that will be used throughout this paper. Further assumptions will be stated when they are needed.

**A1** We observe i.i.d data $(X_i, Y_i), i = 1, \ldots, n + 1$ from a common distribution $P$ on $\mathbb{R}^p \times \mathbb{R}$ with mean function $\mu(x) = E(Y|X = x)$.

**A2** For $(\mathbf{X}, \mathbf{Y}) \sim P$, the noise variable $\epsilon_i = Y_i - \mu(X_i)$ is independent of $X_i$, and the density function of $\epsilon$ is symmetric about 0 and nonincreasing on $\mathbb{R}_+$.

**A3** The density function of $\epsilon$ is bounded away from zero by $r > 0$ in a neighborhood of its $\alpha$ upper quantile.

Assumption A1 is a common assumption in the regression literature. Assumption A2 is less stringent than the assumptions typically found in the statistical literature, as it does not necessitate that $\epsilon$ has a finite first moment. Furthermore, the symmetry and monotonicity conditions can be relaxed by considering appropriate quantiles or density level sets of $\epsilon$; see more details in Lei et al. (2018). Assumption A3 is crucial for ensuring that the estimator of the $\alpha$ upper quantile of $\alpha$ is close to its true value, which is essential for the proof. Specifically, the quantile function of $\epsilon$ satisfies $\gamma$-Hölder continuity at its $\alpha$ upper quantile with $\gamma = 1$; see lemma 1.

Inspired by Lei et al. (2018), to quantify the efficiency of the prediction bands, we compare its length to the idealized prediction band. Our work focuses on the linear regression model, where we denote $\mu(x) = x\boldsymbol{\beta}$ with the parameter vector $\boldsymbol{\beta}$ is $p$-dimensional. The estimator of the prediction is represented as $\hat{\mu}_n(x)$. The oracle prediction band is defined as

$$C_s^*(x) = [\mu(x) - q_{1-\alpha}, \mu(x) + q_{1-\alpha}],$$

where $q_{1-\alpha}$ is the $\alpha$ upper quantile of $\mathcal{L}(|\epsilon|)$. This band assumes complete knowledge of the regression function $\mu(x)$ and the error distribution. Under Assumptions A1 and A2, the band is optimal in the sense outlined in Lei et al. (2018):

- it is has valid conditional coverage: $\mathbb{P}(Y \in C(x) \mid X = x) \geq 1 - \alpha$;

- it has the shortest length among all bands with conditional coverage;

- it has the shortest average length among all bands with marginal coverage.

### 2.2 CONFORMAL PREDICTION

Let $(X_i, Y_i) \in \mathbb{R}^p \times \mathbb{R}, i = 1, ..., n$ denote training samples. Given a desired coverage rate $\alpha \in (0, 1)$, conformal prediction constructs a prediction band $\hat{C}_n : \mathbb{R}^p \to \{\text{subsets of } \mathbb{R}\}$ for $Y_{n+1}$ at a test point $X_{n+1}$ satsifying $P(Y_{n+1} \in \hat{C}_n(X_{n+1}) \geq 1 - \alpha$, under the assumption that all pairs $(X_i, Y_i)_{i=1}^{n+1}$ are exchangeable. The idea behind the method is extremely simple: for each $y \in \mathbb{R}$, we test if $y$ is plausible value for $Y_{n+1}$ given $(X_i, Y_i)_{i=1}^n$ and $X_{n+1}$ such that $(X_i, Y_i)_{i=1}^{n+1}$ look like exchangeable data. Since conformal prediction only relies on the assumption of exchangeability, it is a flexible approach that can be applied using various algorithms, including those for regression, classification, and unsupervised settings such as clustering and principal components analysis. In this paper, we focus specifically on regression models.

Given a model $\hat{\mu}^y : \mathcal{X} \to \mathbb{R}$ that was fitted on the dataset $(X_i, Y_i)_{i=1}^n \cup (X_{n+1}, y)$, for each $y \in \mathbb{R}$ we define nonconformity score function:

$$R_i^y = \begin{cases} |Y_i - \hat{\mu}^y(X_i)|, & i = 1, \ldots, n \\ |y - \hat{\mu}^y(X_{n+1})|, & i = n + 1. \end{cases} \tag{2}$$

The nonconformity score function typically involves a model-fitting process that evaluates the degree of agreement between the latest input and the fitted model. A lower nonconformity score indicates a higher concordance between the fitted model and the sample model. It is important to note that the nonconformity score function is not unique. For example, Lei et al. (2018) constructed a standardized absolute fitted residual

$$R_i^y = |Y_i - \hat{\mu}(X_i)| / \hat{\sigma}(x), i \in \mathcal{I}_2$$

where the conditional mean $\hat{\mu}$ and conditional MAD $\hat{\sigma}(x)$ are fit on samples in training dataset $\mathcal{I}_1$, and $\mathcal{I}_2$ denotes the validation set. Similar improvements using quantile regression occur in Kivaranovic et al. (2020) and Romano et al. (2019). We note that these improvements are based on split conformal prediction, while this paper focuses on full conformal prediction. After that, we rank $R_{n+1}^y$ among the fitted residual $R_1^y, \ldots, R_{n+1}^y$, and compute $p$-value:

$$\hat{p}^y = \frac{1}{n+1} \sum_{i=1}^{n+1} \delta_{\{R_i^y \leq R_{n+1}^y\}}, \tag{3}$$

where $\delta$ is the indicator function. Then the prediction interval at $X_{n+1}$ is obtained by thresholding the $p$-value:

$$\hat{C}_n(X_{n+1}) = \{y : \hat{p}^y \geq \alpha\}. \tag{4}$$

Equivalently, we can write $\hat{C}_n(X_{n+1})$ as

$$\hat{\mu}(X_{n+1}) \pm (\text{the} \lceil (1 - \alpha)(n + 1) \rceil \text{-th smallest of} (R_1^y, \ldots, R_{n+1}^y)). \tag{5}$$

The conformal method is well-known to have finite sample and distribution-free coverage:

**Proposition 1** (Vovk et al. (2005)). *If $(X_i, Y_i), i = 1, \ldots, n$ are i.i.d., then for an new i.i.d. pair $(X_{n+1}, Y_{n+1})$, we have*

$$P\left(Y_{n+1} \in \hat{C}_n(X_{n+1})\right) \geq 1 - \alpha,$$

*If we assume additionally that for all $y \in \mathbb{R}$, the fitted absolute residuals $\{R_i = |Y_i - X_i\hat{\theta}|\}_{i=1}^{n+1}$ have a continuous joint distribution, then it also holds that*

$$P(Y_{n+1} \in \hat{C}_n(X_{n+1})) \leq 1 - \alpha + \frac{1}{n+1}.$$

We note that the step (2) and step (3) must be repeated each time when producing a prediction interval, which is impossible in practice. Therefore, we often use a discrete grid of trail values $y$ or use homotopy methods.

## 3 CONFORMAL PREDICTION WITH MODEL-AWARE DEBIASING

Recall the conformal prediction band constructed in Section 2.2, the width of band is $2T_{1-\alpha}(|Y_i - \hat{Y}_i|)$ in 5, where $T_{1-\alpha}(|Y_i - \hat{Y}_i|)$ is $\lceil (1 - \alpha)(n + 1) \rceil$-th quantile of $(n + 1)^{-1} \sum_{i=1}^{n+1} \delta_{\{|Y_i - \hat{Y}_i| \leq t\}}$. A natural thought is that if $\hat{Y}_i$ is closer to the ground truth of $Y_i$, the corresponding version of nonconformity scores is likely to decrease, which may result in a narrower prediction band. More specifically, we can alleviate the bias when constructing the nonconformity score function. We make a small experiment on simulated data to show how it improves upon standard conformal prediction in Fig.1.

Most models produce unavoidably biased solutions especially in high-dimensional settings, since a point estimate $\hat{\theta} \in \mathbb{R}^p$ must be produced from data in lower dimension. Take ridge regression as an example, suppose the parameter of interest is $\mathbf{a}^T\boldsymbol{\beta}$ in a linear model $\mathbf{y} = \mathbf{X}\boldsymbol{\beta} + \boldsymbol{\epsilon}$; here,

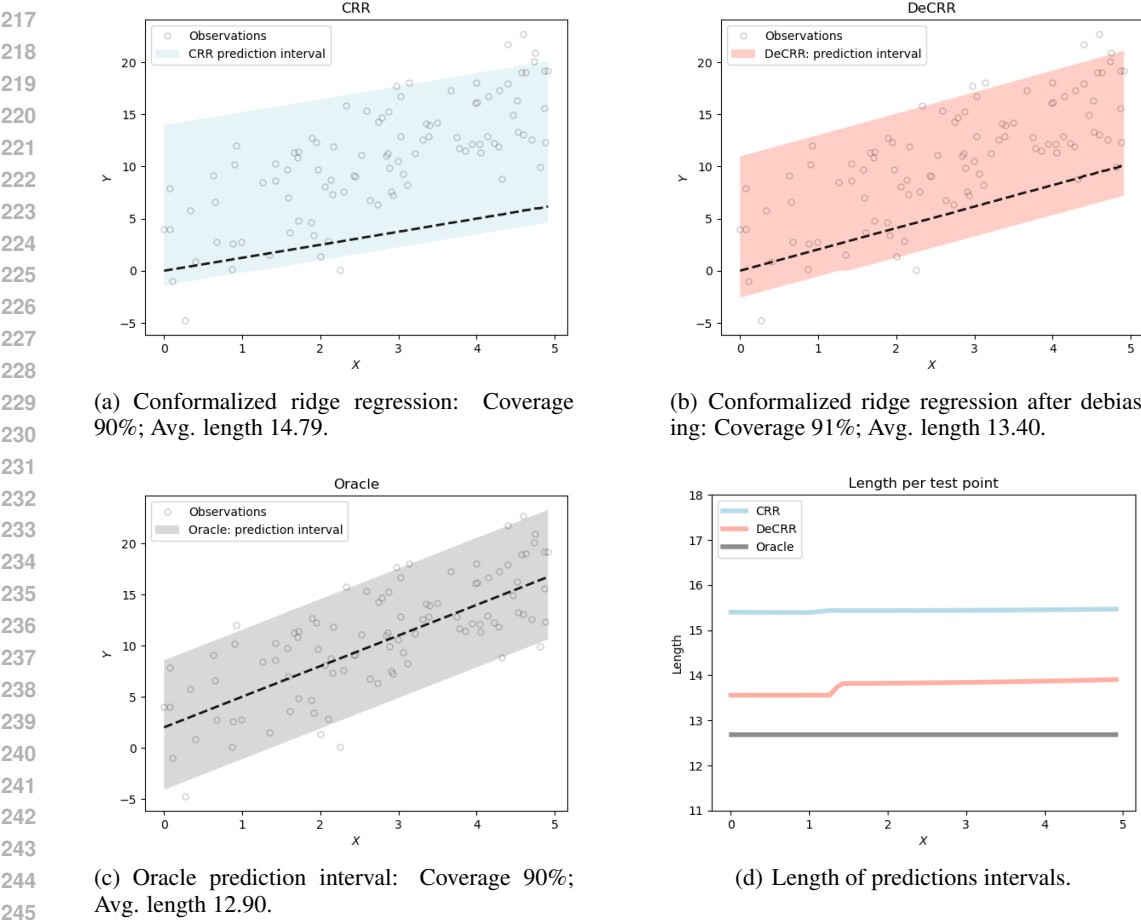

(a) Conformalized ridge regression: Coverage 90%; Avg. length 14.79.

(b) Conformalized ridge regression after debiasing: Coverage 91%; Avg. length 13.40.

(c) Oracle prediction interval: Coverage 90%; Avg. length 12.90.

(d) Length of predictions intervals.

Figure 1: Prediction intervals on simulated data with outliers: (a) the standard conformalized ridge regression, (b) conformalized ridge regression after debiasing, and (c) oracle prediction interval. The length of the interval as a function of X is shown in (d). The target coverage rate is 90%. The broken black curve in (a), (b) and (c) is the pointwise prediction from the ridge regression.

the dimension $p < n$, $\mathbf{X}$ has rank $p$, and $\boldsymbol{a}$ is a known vector. The ridge estimator is $\boldsymbol{a}^T\hat{\boldsymbol{\theta}}$ with $\hat{\boldsymbol{\theta}} = \left(\mathbf{X}^T\mathbf{X} + \rho_n\mathbf{I}_p\right)^{-1}\mathbf{X}^T\mathbf{y}$, for some $\rho_n > 0$, with $\mathbf{I}_p$ denoting the $p$-dimensional identity matrix. Perform a thin singular value decomposition $\mathbf{X} = \mathbf{P}\boldsymbol{\Lambda}\mathbf{Q}^T$ as in Theorem 7.3.2 in Horn & Johnson (2012), where $\mathbf{P}$ and $\mathbf{Q}$ is $n \times p$ and $p \times p$ orthonormal matrices and $\Lambda$ is an $p \times p$ diagonal matrix of full rank. Assume the error vector $\boldsymbol{\epsilon}$ consists of independent identically distributed (i.i.d.) components. Then the bias and the standard deviation can be calculated (and controlled) as follows:

$$\mathrm{E}\,\mathbf{a}^T\hat{\boldsymbol{\theta}} - \mathbf{a}^T\boldsymbol{\beta} = -h_n\mathbf{a}^T\mathbf{Q}\left(\boldsymbol{\Lambda}^2 + h_n\mathbf{I}_p\right)^{-1}\mathbf{Q}^T\boldsymbol{\beta}$$

which implies

$$\left|\mathrm{E}\,\mathbf{a}^T\hat{\boldsymbol{\theta}} - \mathbf{a}^T\boldsymbol{\beta}\right| \leq \frac{\rho_n\|\mathbf{a}\|_2 \times \|\boldsymbol{\beta}\|_2}{\lambda_p^2 + \rho_n},$$

$$\sqrt{\mathrm{Var}\left(\mathbf{a}^T\hat{\boldsymbol{\theta}}\right)} = \sqrt{\mathrm{Var}\left(\epsilon_1\right) \times \mathbf{a}^T\boldsymbol{Q}\left(\boldsymbol{\Lambda}^2 + \rho_n\mathbf{I}_p\right)^{-2}\boldsymbol{\Lambda}^2\boldsymbol{Q}^T\mathbf{a}}$$

$$\leq \frac{\sqrt{\mathrm{Var}\left(\epsilon_1\right)} \times \|\mathbf{a}\|_2}{\lambda_p}.$$

If $\|\boldsymbol{\beta}\|_2$ does not have a bounded order, the bias is significantly larger than the standard deviation, and may tend to infinity which makes prediction interval difficult (Zhang & Politis (2022)). To

address this issue, we account for the bias when constructing the nonconformity score function:

$$\tilde{R}_i^y = |Y_i - \hat{\mu}(X_i) + bias(\hat{\mu}(X_i))|, \quad i = 1, ..., n,$$
$$\tilde{R}_{n+1}^y = |y - \hat{\mu}(X_{n+1}) + bias(\hat{\mu}(X_{n+1}))|. \tag{6}$$

And the prediction band $\hat{C}_n(X_{n+1})$ is

$$\left\{ y \in \tilde{\mathbb{R}} : \tilde{R}_{n+1}^y \leq Q_{1-\alpha}\left( \sum_{i=1}^n \frac{1}{n+1} \cdot \delta_{\tilde{R}_i^y} + \frac{1}{n+1} \cdot \delta_{+\infty} \right) \right\}.$$

The following result shows that the conformal prediction band with model-aware debiasing retains finite sample validity.

**Theorem 1.** *If* $(X_i, Y_i)$, $i = 1, \ldots, n$ *are i.i.d., then for a new i.i.d. pair* $(X_{n+1}, Y_{n+1})$.

$$\mathbb{P}\left( Y_{n+1} \in \hat{C}^{Debias}(X_{n+1}) \right) \geq 1 - \alpha.$$

*If we assume additionally that for all* $y \in \mathbb{R}$*, the fitted absolute residuals* $\{\tilde{R}_i\}_{i=1}^{n+1}$ *have a continuous joint distribution, then it also holds that*

$$P\left( Y_{n+1} \in \hat{C}^{Debias}(X_{n+1}) \right) \leq 1 - \alpha + \frac{1}{n+1}.$$

The bias-correction step of the nonconformity score function remains the exchangeability within training data and test data. Therefore, the proof of this theorem is similar to the classical conformal prediction, and we omit it here. We note that the bias is usually unknown unfortunately. So it is often replaced by its estimation in practice.

### 3.1 CONFORMALIZE THRESHOLD RIDGE REGRESSION WITH MODEL-AWARE DEBIASING

In high-dimensional models, linear regression is the most common example. As noted by Zhang & Politis (2022) and Shao & Deng (2012), threshold ridge regression is computationally much simpler than methods such as the LASSO(Tibshirani (1996)), SCAD(Fan & Li (2001)) and the ENET(Zou & Hastie (2005)) and may be preferable in some settings. Based on the above discussion, we give results about coverage guarantee and efficiency on threshold ridge regression in this section. It reduces to the classical ridge regression model as the threshold parameter approaches zero. Conformal prediction can be easily applied to the ridge regression model using the homotopy method described by Vovk et al. (2005), and its efficiency has been well-studied by Burnaev & Vovk (2014). The procedure for applying conformalized threshold ridge regression with model-aware debiasing is summarized in Algorithm 1.

Perform a thin singular value decomposition $\mathbf{X} = \mathbf{P}\mathbf{\Lambda}\mathbf{Q}^\top$ as before, where $\mathbf{P}$ and $\mathbf{Q}$ is $n \times r$ and $p \times r$ orthonormal matrices, and $\Lambda$ is an $r \times r$ diagonal matrix. Denote $\mathbf{Q}_\perp$ as the $p \times (p-r)$ orthonormal complement of $\mathbf{Q}$, which satisfies the following properties:

$$\mathbf{Q}_\perp^T \mathbf{Q}_\perp = \mathbf{I}_{p-r}, \quad \mathbf{Q}^T \mathbf{Q}_\perp = 0 \quad \text{and} \quad \mathbf{Q}\mathbf{Q}^T + \mathbf{Q}_\perp \mathbf{Q}_\perp^T = \mathbf{I}_p.$$

Here, the matrix of zeros is of appropriate dimensions. Define $\boldsymbol{\theta} = \mathbf{Q}\mathbf{Q}^\top \boldsymbol{\beta}$ and $\boldsymbol{\theta}_\perp = \mathbf{Q}_\perp \mathbf{Q}_\perp^T \boldsymbol{\beta}$, so $\boldsymbol{\beta} = \boldsymbol{\theta} + \boldsymbol{\theta}_\perp$. According to Shao & Deng (2012), the ridge regression estimate $\boldsymbol{\theta}$ rather than $\boldsymbol{\beta}$. If the design matrix $\mathbf{X}$ has rank $p \leq n$, then $\mathbf{Q}_\perp$ does not exist and we set $\boldsymbol{\theta}_\perp = 0$ in this case. For a chosen ridge parameter $h_n > 0$, we define the classical ridge regression estimator as $\hat{\boldsymbol{\theta}} = (\mathbf{X}^T\mathbf{X} + h_n\mathbf{I_p})^{-1}\mathbf{X}^T\mathbf{y}$. For a threshold $a_n$, we define the set and the estimator $\tilde{\theta}_i$ as follows:

$$\mathcal{M}_{a_n} = \{i \mid |\theta_i| > a_n\}, \quad \tilde{\theta}_i = \hat{\theta}_i \times \mathbf{1}_{i \in \mathcal{M}_{a_n}}. \tag{7}$$

Let $q_n$ denote the number of elements in the set $\mathcal{M}_{a_n}$. We define $c_{ik} = \sum_{j \in \mathcal{M}_{a_n}} x_{ij} q_{jk}, \forall i = 1, ..., n, k = 1, ..., r$, where $\mathbf{Q} = (q_{jk})_{j=1,...,n,k=1,...,r}$. In the threshold ridge regression model, we define threshold estimator $\hat{\boldsymbol{\theta}}^*$ by letting the components of $\hat{\boldsymbol{\theta}}$ whose absolute value less than or equal to the threshold parameter $a_n$ be zero.

Besides the conditions in Section 2.2, we need some additional assumptions.

---

**Algorithm 1** Conformalize Threshold Ridge Regression with Model-Aware Debiasing

---

**Input:**

Data $(X_i, Y_i), i = 1, ..., n$, prescribed error level $\alpha$, threshold parameter $a_n$ and ridge parameter $h_n$, points $\mathcal{X}_{\text{new}} = \{X_{n+1}, X_{n+2}, ...\}$ which are to construct prediction bands

**Output:**

Prediction bands at each points in $\mathcal{X}_{\text{new}}$

1: **for** $x \in \mathcal{X}_{\text{new}}$ **do**
2:      Set $\hat{\boldsymbol{\theta}} = (\mathbf{X}^\top \mathbf{X} + h_n \mathbf{I}_p)^{-1} \mathbf{X}^\top \mathbf{y}$ and calculate $\mathcal{M}_{a_n}$
3:      Calulate the hat matrix $\mathbf{H} := \mathbf{P}\boldsymbol{\Lambda}[(\boldsymbol{\Lambda}^2 + h_n \mathbf{I}_r)^{-1} + h_n(\boldsymbol{\Lambda}^2 + h_n \mathbf{I}_r)^{-2}]\boldsymbol{\Lambda}\mathbf{P}^\top$, where the elements in $i$-th row are zeros, $i \in \mathcal{M}_{a_n}$
4:      Set $\mathbf{A} = (a_1, ..., a_{n+1})^\top := (\mathbf{I}_{n+1} - \mathbf{H})(y_1, ..., y_n, 0)^\top$
5:      Set $\mathbf{B} = (b_1, ..., b_{n+1})^\top := (\mathbf{I}_{n+1} - \mathbf{H})(0, ..., 0, 1)^\top$
6:      **for** $i = 1, ..., n$ **do**
7:        **if** $b_{n+1} - b_i > 0$ **then**
8:          set $u_i := l_i := (a_i - a_{n+1})/(b_{n+1} - b_i)$
9:        **else**
10:          set $u_i := \infty$ and $l_i := -\infty$
11:        **end if**
12:      **end for**
13:      sort $u_1, ..., u_n$ and $l_1, ..., l_n$ in the ascending order obtaining $u_{(1)}, ..., u_{(n)}$ and $l_{(1)}, ..., l_{(n)}$
       $\hat{C}_n(x) = [l_{(\lfloor (\alpha/2)(n+1) \rfloor)}, u_{(\lceil (1-\alpha/2)(n+1) \rceil)}]$
14: **end for**
15: **return** $\hat{C}_n(x)$ for each points in $\mathcal{X}_{\text{new}}$

---

**B1** The largest positive eigenvalue $\lambda_1$ and smallest positive eigenvalue $\lambda_r$ of $\mathbf{X}^\top \mathbf{X}$, satisfies
$$\lambda_r = O(n^\eta), \quad 0 < \eta \le 1 \text{ and } \eta \text{ does not depend on } n.$$

**B2** We assume that
$$\|\boldsymbol{\theta}\| = O(n^{\alpha_\theta}), \quad 0 < \alpha_\theta < 4\eta \text{ and } \alpha_\theta \text{ does not depend on } n.$$

**B3** The $\varepsilon_i$ is assumed that
$$E|\varepsilon_i|^m < \infty \quad \text{for an even integer } m \text{ not depending on } n.$$

**B4** The dimension is assumed as
$$p = O(n^{\alpha_p}), \alpha_p \text{ does not depend on } n.$$

**B5** We assume the ridge parameter that
$$h_n = O(n^{2\eta - \delta}), \quad \alpha_\theta - \eta < \delta \text{ and } \frac{\eta + \alpha_\theta}{2} < \delta.$$

**B6** We assume the threshold parameter that
$$a_n = O(n^{-\alpha_a}), \alpha_a > 0 \text{ and } \alpha_a + \frac{\alpha_p}{m} - \eta < 0.$$
Furthermore, we assume $\exists$ a constant $0 < c_a < 1$ such that $\max_{i \notin \mathcal{M}_{a_n}} |\theta_i| \le c_a \times a_n$, and $\min_{i \in \mathcal{M}_{a_n}} |\theta_i| \ge \frac{a_n}{c_a}$.

**B7** We assume there exists a constant $C_{\alpha_\mathcal{N}} > 0$ such that $\sum_{k=1}^r c_{ik}^2 \le C_{\alpha_\mathcal{N}}$.

These assumptions are common in the statistical literature. Assumption B1 guarantees that the smallest positive eigenvalue of $\mathbf{X}^\top \mathbf{X}$ ensures the covariance matrix is well-conditioned and the regression problem is stable with respect to small perturbations in the data. Examples satisfying the condition are provided Bai & Yin (2008). In high-dimensional regression contexts, the sparsity of $\boldsymbol{\beta}$ implies the sparsity of $\boldsymbol{\theta}$; thus, we impose a sparsity condition on $\boldsymbol{\theta}$ as outlined in Assumption B2. Assumption B3 ensures that our estimators are robust to extreme values, facilitating consistent statistical inference even in high-dimensional or small-sample settings. Furthermore, this assumption can be substituted with a normality condition, which is a specific case of the broader assumption. Assumption B4 requires that the dimension of the parameter vector $p$ diverges at a polynomial rate, which can be much larger than $n$. Assumptions B5, B6, and B7 are similar to the conditions presented in Zhang & Politis (2022), but they are formulated in a more relaxed manner here. We note that these assumptions impose restrictions on the estimators obtained from threshold ridge regression. Even if they are violated, the prediction intervals derived from our method still satisfy marginal validity.

The following result shows the efficiency of prediction intervals produced by the standard conformal prediction in threshold ridge regression.

**Theorem 2.** *Fix $\alpha \in (0,1)$, and let $\hat{C}_{ThCRR}$ denote the conformal interval of the threshold ridge regression. Under assumptions A1-A3 and B1-B7, we have*

$$\text{Width}\left(\hat{C}_{ThCRR}(X_{n+1})\right) - 2q_{1-\alpha} = O_p(n^{\alpha_\theta - \delta}). \tag{8}$$

Theorem 2 demonstrates that the conformal interval converges to the oracle prediction band if $\alpha_\theta < \delta$ as $n \to \infty$. Unfortunately, $\alpha_\theta$ is typically not greater than $\delta$, and thus the efficiency of the conformal interval generally lacks theoretical guarantees without stringent assumptions, such as sampling stability, perturbation sensitivity, and consistency of the base estimator, as discussed in Lei et al. (2018).

To mitigate the bias of the estimator, we define the debiased nonconformity score function as follows:

$$\begin{aligned}
\tilde{R}_i^y &= |Y_i - X_i\hat{\boldsymbol{\theta}}^* - h_n X_i \boldsymbol{Q}(\boldsymbol{\Lambda}^2 + h_n \mathbf{I}_r)^{-1}\mathbf{Q}^\top \hat{\boldsymbol{\theta}}^*|, \quad i = 1, ..., n, \\
\tilde{R}_{n+1}^y &= |y - X_{n+1}\hat{\boldsymbol{\theta}}^* - h_n X_{n+1}\mathbf{Q}(\boldsymbol{\Lambda}^2 + h_n \mathbf{I}_r)^{-1}\mathbf{Q}^\top \hat{\boldsymbol{\theta}}^*|.
\end{aligned} \tag{9}$$

And the prediction band $\hat{C}_{\text{ThCRR}}^{\text{Debias}}(X_{n+1})$ is $(X_{n+1}\hat{\boldsymbol{\theta}}^* - h_n X_{n+1}\mathbf{Q}(\boldsymbol{\Lambda}^2 + h_n \mathbf{I}_r)^{-1}\mathbf{Q}^\top \hat{\boldsymbol{\theta}}^* \pm$ the $\lceil(1-\alpha)(n+1)\rceil$-th smallest of $(\tilde{R}_1^y, ..., \tilde{R}_{n+1}^y))$. The efficiency of the intervals generated by conformal prediction with model-aware debiasing in threshold ridge regression is outlined as follows:

**Theorem 3.** *Fix $\alpha \in (0,1)$, and let $\hat{C}_{ThCRR}^{Debias}$ denote the conformal interval through model-aware debiasing in the threshold ridge regression. Under the same conditions as in Theorem 2, we have*

$$\text{Width}\left(\hat{C}_{ThCRR}^{Debias}(X_{n+1})\right) - 2q_{1-\alpha} = O_p(n^{-\eta}). \tag{10}$$

Since $\eta$ is usually positive, the interval produced by our method converges to the oracle prediction interval in Lei et al. (2018) at a certain rate, whereas the classical conformal interval may not. The proof of Theorem 3 is presented in Appendix B.

Finally, we return to the discussion of the validity of prediction intervals. While the marginal validity is established by Theorem 1, we seek stronger assurances. To this end, we leverage the definition of asymptotic conditional validity from Lei et al. (2018) to demonstrate that our proposed method possesses asymptotic conditional validity under certain conditions.

**Definition 1.** *We say that prediction bands have asymptotic conditional coverage at the level $(1-\alpha)$ if there exist random sets $\Lambda_n \subseteq \mathbb{R}^d$ such that $P(X_{n+1} \in \Lambda_n \mid \Lambda_n) = 1 - o_p(1)$ and*

$$\sup_{X_{n+1} \in \Lambda_n} |\mathbb{P}(Y \in C_n(X_{n+1}) \mid X_{n+1} = x_{n+1}) - (1-\alpha)| = o_p(1)$$

The following theorem shows that the prediction interval produced by our method has asymptotic conditional coverage at the level $1 - \alpha$.

**Theorem 4.** *Under the same condition in Theorem 3, we have*

$$L(\hat{C}_{ThCRR}^{Debias}(X_{n+1}) \triangle C_s^*(x)) = o_p(1) \tag{11}$$

*where $L(\cdot)$ denotes the Lebesgue measure and $\triangle$ denotes the symmetric difference between two sets.*

## 4 NUMERICAL SIMULATIONS

In this section, we systematically compare the conformalize threshold ridge regression with model-aware debiasing with the standard conformal prediction version and the hybrid bootstrap method Zhang & Politis (2022), focusing on different $p/n$ ratio. We conduct experiments on five benchmark datasets for the case where $n > p$: facebook_1($n$=754, $p$=53), facebook_2($n$=814, $p$=53), bio($n$=458, $p$=9), bike($n$=1089, $p$=18) and concrete($n$=510, $p$=8). For the scenario where $n < p$, we perform experiments on two additional benchmark datasets. These datasets were previously also considered by Romano et al. (2019).

For ease of interpretation, we center and scale the features to achieve zero mean and unit variance, while scaling the response variables by dividing them by their mean absolute value. We report four key metrics: the average coverage, the average length of the prediction set, and their respective standard errors. In addition, we measure conditional coverage and results are shown in Appendix C. These performance metrics are averaged over 20 different training-test splits, with $90\%$ of the data used for training and the remaining $10\%$ reserved for testing. Throughout the experiments, the nominal miscoverage rate is fixed at $\alpha = 0.1$. In our data examples, the optimal ridge parameter $h_n$ and threshold $a_n$ are determined by 5-fold cross-validation.

Table 1: Comparison of prediction intervals when $n < p$. The nominal miscoverage rate is fixed at $\alpha = 0.1$. 'CRR' abbreviates 'Conformalized threshold ridge regression", "DeCRR" abbreviates 'Conformalized threshold ridge regression with model-aware debiasing', and 'Boot_DeRR' abbreviates Bootstrap method in debiased threshold ridge regression. The standard errors are in parantheses.

| Case | Dataset | Method | Coverage | **Length** |
|---|---|---|---|---|
| $n > p$ | facebook_1 | CRR | 0.932(0.008) | 4.262(0.177) |
| | | DeCRR | 0.921(0.0145) | **4.121(0.219)** |
| | | Boot_DeRR | 0.902(0.000) | 7.152(0.397) |
| | facebook_2 | CRR | 0.885(0.006) | 4.174(0.259) |
| | | DeCRR | 0.876(0.012) | **4.100(0.299)** |
| | | Boot_DeRR | 1.000(0.000) | 202.834(5.574) |
| | bio | CRR | 0.978(0.001) | 2.195(0.049) |
| | | DeCRR | 0.967(0.010) | **2.118(0.096)** |
| | | Boot_DeRR | 0.660(0.010) | 2.926(0.058) |
| | bike | CRR | 0.900(0.011) | 2.452(0.010) |
| | | DeCRR | 0.900(0.005) | **2.439(0.014)** |
| | | Boot_DeRR | 0.812(0.012) | 3.056(0.095) |
| | concrete | CRR | 0.912(0.020) | 0.962(0.0128) |
| | | DeCRR | 0.913(0.015) | **0.958(0.011)** |
| | | Boot_DeRR | 0.461(0.041) | 1.795(0.033) |

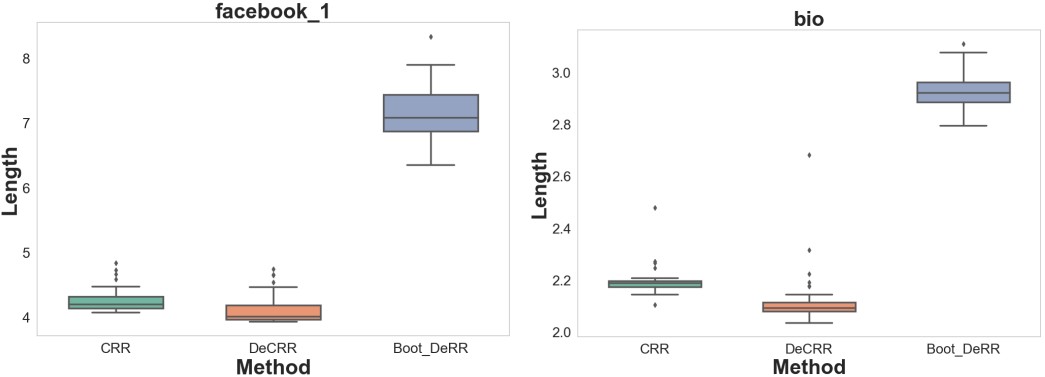

Figure 2: Length of prediction intervals on the *facebook_1* and *bio* datasets when $n > p$.

Table 1 and Fig 2 summarize the results of the first case. The average coverage of prediction intervals produced by the bootstrap method in Zhang & Politis (2022) is notably high, sometimes reaching up to one in the facebook_2 dataset. However, this comes at the expense of a larger interval length, as the intervals generated by this method are significantly wider than those produced by conformal methods, particularly in the facebook_2 dataset, rendering them nearly impractical. Additionally, the coverage exhibits high variability, which is undesirable in practical applications.

In contrast, the coverage of the conformal intervals is more stable and approaches the target value of $90\%$. Furthermore, the experiments consistently demonstrate that, on the one hand, conformal methods yield shorter intervals compared to the bootstrap method. On the other hand, following the

bias correction step, the intervals become even shorter than a standard conformal prediction, thereby confirming that our method enhances the efficiency of the prediction intervals while retaining finite-sample coverage.

Table 2: Comparison of prediction intervals when $n < p$.

| Case | Dataset | Method | Coverage | Length |
|---|---|---|---|---|
| | | CRR | 0.993(0.025) | 2.274(0.054) |
| | community | DeCRR | 0.987(0.034) | **2.152(0.114)** |
| | | Boot_DeRR | 1.000(0.000) | 3.577(0.092) |
| $n < p$ | blog_data | CRR | 0.933(0.001) | 3.206(0.326) |
| | | DeCRR | 0.933(0.001) | **2.901(0.336)** |
| | | Boot_DeRR | 0.934(0.001) | 8.197(0.186) |

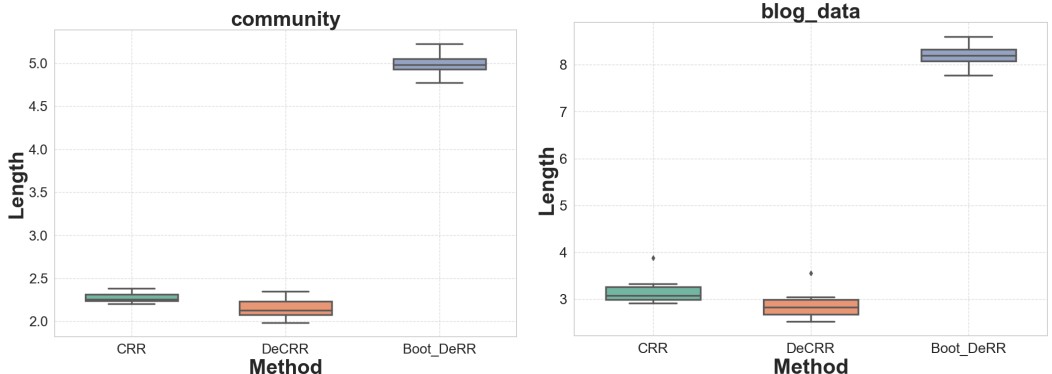

Figure 3: Prediction intervals on two benchmark datasets when $n < p$.

Table 2 and Fig 3 summarize the results of the second case. A similar observation is that, on average, our method produces shorter prediction intervals compared to both the standard conformal prediction and the bootstrap method while successfully constructing prediction bands at the nominal coverage rate of $90\%$.

## 5 CONCLUSION

In this paper, we propose conformal prediction with model-aware debiasing as a novel approach that leverages known information from the data to formulate the debiasing nonconformity score function and leaves the unknown aspects to the conformal prediction framework. Our method enhances the efficiency of prediction intervals while preserving finite-sample coverage. Notably, it achieves stronger validity, including asymptotic conditional coverage under some conditions.

We plan to extend this concept to more general settings. Such an extension of conformal prediction holds promise not only for regression problems but also for classification and other unsupervised learning tasks, such as clustering. Specifically, we aim to enhance our method to estimate a predictive probability distribution (see more details in Izbicki et al. (2022)) using a bias correction approach, rather than merely providing interval estimates. This is particularly important since oracle prediction intervals can often be quite large, especially in the context of mixed models. Therefore, we propose considering the highest predictive density set as the oracle prediction band. This predictive set requires minimal assumptions and represents the smallest Lebesgue measure with local validity and asymptotic conditional validity, thereby facilitating improved performance in a broader range of scenarios.

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

## A    LEMMAS AND PROOFS

**Lemma 1.** *If the density of $|Y - X\boldsymbol{\theta}|$ is bounded below by $l > 0$ in a neighborhood of its $\alpha$ upper quantile, then $F_\theta^{-1}$ is Hölder continuous on this neigborhood with $\gamma$-Hölder constant $1/l$ and $\gamma = 1$.*

*Proof.* Assume the density of $|Y - X\boldsymbol{\theta}|$ is bounded from below from by $l > 0$ in a neighborhood of its $\alpha$ upper quantile $\left[ F_\theta^{-1}\left(q_\alpha(\boldsymbol{\theta}) - l^*\right), F_\theta^{-1}\left(q_\alpha(\boldsymbol{\theta}) + l^*\right) \right]$ for some $l^* > 0$, then for any $q_1, q_2 \in [q_\alpha(\boldsymbol{\theta}) - l^*, q_\alpha(\boldsymbol{\theta}) + l^*]$, assume WLOG that $q_2 \geq q_1$,

$$
\begin{aligned}
F_\theta^{-1}(q_2) - F_\theta^{-1}(q_1) &= \{t_2 - t_1 : \mathbb{P}\left(|Y - X\boldsymbol{\theta}| \leq t_2\right) = q_2, \mathbb{P}\left(|Y - X\boldsymbol{\theta}| \leq t_1\right) = q_1\} \\
&\leq \{t_2 - t_1 : \mathbb{P}\left(|Y - X\boldsymbol{\theta}| \in (t_1, t_2]\right) = q_2 - q_1\} \\
&\leq \{t_2 - t_1 : l\left(t_2 - t_1\right) \leq \mathbb{P}\left(|Y - X\boldsymbol{\theta}| \in (t_1, t_2]\right) = q_2 - q_1\} \\
&\leq (q_2 - q_1)/l.
\end{aligned}
$$

$\square$

**Lemma 2.** *For two cumulative distribution functions $F_1$ and $F_2$, set*

$$
\Delta := \sup_t |F_1(t) - F_2(t)|.
$$

*If $F_1^{-1}(\cdot)$, $F_2^{-1}(\cdot)$ exist, and $F_2^{-1}(\cdot)$ is $\gamma$-Hölder continuous on $[q - \Delta, q + \Delta]$ for $\gamma \in (0, 1)$, then it holds that*

$$
\left| F_1^{-1}(q) - F_2^{-1}(q) \right| \leq \mathfrak{L}\Delta^\gamma,
$$

*where $\mathfrak{L}$ is the Hölder continuity constant.*

*Proof.* Note that

$$
\begin{aligned}
&\left| F_1\left(F_1^{-1}(q)\right) - F_2\left(F_1^{-1}(q)\right) \right| \leq \Delta \\
\Rightarrow &q - \Delta \leq F_2\left(F_1^{-1}(q)\right) \leq q + \Delta \\
\Rightarrow &F_2^{-1}(q - \Delta) \leq F_1^{-1}(q) \leq F_2^{-1}(q + \Delta).
\end{aligned}
$$

Therefore, using the Hölder continuity assumption, we obtain

$$
F_1^{-1}(q) - F_2^{-1}(q) \leq F_2^{-1}(q + \Delta) - F_2^{-1}(q) \leq \mathfrak{L}\Delta^\gamma
$$

and

$$
F_1^{-1}(q) - F_2^{-1}(q) \geq F_2^{-1}(q - \Delta) - F_2^{-1}(q) \geq -\mathfrak{L}\Delta^\gamma.
$$

Hence the proof is completed. $\square$

**Lemma 3.** *Denote $F_n$ is the empirical CDF of $|Y_i - X_i\boldsymbol{\theta}|$, and $F_{\hat{n}}$ is the empirical CDF of $|Y_i - X_i\hat{\boldsymbol{\theta}}|$. On the event $\{|X_i\hat{\boldsymbol{\theta}} - X_i\boldsymbol{\theta}| \leq \rho_n\}$, we have*

$$
|F_n^{-1}(t) - F_{\hat{n}}^{-1}(t)| \leq \rho_n, \quad \forall t \in [0, 1].
$$

*Proof.* On the event $\{|X_i\hat{\boldsymbol{\theta}} - X_i\boldsymbol{\theta}| \leq \rho_n\}$, we have $|Y_i - X_i\boldsymbol{\theta}| - |Y_i - X_i\hat{\boldsymbol{\theta}}| \leq \rho_n$. Therefore according to the definition of the empirical CDF, we have

$$
F_{\hat{n}}(t - \rho_n) \leq F_n(t) \leq F_{\hat{n}}(t + \rho_n).
$$

Assume that $t_1$ and $t_2$ are the $q \in [0, 1]$ quantiles of $F_n(t)$ and $F_{\hat{n}}(t)$ respectively, that is, for $\forall \epsilon > 0$,

$$
\begin{aligned}
F_n(t_1 - \epsilon) &< q, \text{ and} F_n(t_1) \geq q, \\
F_{\hat{n}}(t_2 - \epsilon) &< q, \text{ and} F_n(t_2) \geq q.
\end{aligned}
$$

Since $q \leq F_n(t_1) \leq F_{\hat{n}}(t_1 + \rho_n)$, we have $t_2 \leq t_1 + \rho_n$. Similarly, we have $t_1 \leq t_2 + \rho_n$. Therefore,

$$
|F_n(t)^{-1} - F_{\hat{n}}^{-1}(t)| \leq \rho_n, \quad \forall t \in [0, 1].
$$

$\square$

**Lemma 4.** *Suppose random variables $\epsilon_1, \ldots, \epsilon_n$ are i.i.d., $E\epsilon_1 = 0$, and $\exists$ a constant $m > 0$ such that $E|\epsilon_1|^m < \infty$. In addition suppose the matrix $\Gamma = (\gamma_{ij})_{i=1,2,\ldots,k, j=1,2,\ldots,n}$ satisfies*

$$\max_{i=1,2,\ldots,k} \sum_{j=1}^{n} \gamma_{ij}^2 \leq D, D > 0$$

*Then $\exists$ a constant $E_0$ which only depends on $m$ and $E|\epsilon_1|^m$ such that for $\forall \delta > 0$,*

$$P\left( \max_{i=1,2,\ldots,k} \left| \sum_{j=1}^{n} \gamma_{ij}\epsilon_j \right| > \delta \right) \leq \frac{kED^{m/2}}{\delta^m}$$

*Proof.* From theorem 2 in Whittle (1960), for any $i = 1, 2, \ldots, k$,

$$P\left( \left| \sum_{j=1}^{n} \gamma_{ij}\epsilon_j \right| > \delta \right) \leq \frac{E\left| \sum_{j=1}^{n} \gamma_{ij}\epsilon_j \right|^m}{\delta^m} \leq \frac{2^m C(m) E|\epsilon_1|^m \left( \sum_{j=1}^{n} \gamma_{ij}^2 \right)^{m/2}}{\delta^m} \leq \frac{2^m C(m) E|\epsilon_1|^m D^{m/2}}{\delta^m}$$

Choose $E_0 = 2^m C(m) E|\epsilon_1|^m$

$$P\left( \max_{i=1,2,\ldots,k} \left| \sum_{j=1}^{n} \gamma_{ij}\epsilon_j \right| > \delta \right) \leq \sum_{i=1}^{k} P\left( \left| \sum_{j=1}^{n} \gamma_{ij}\epsilon_j \right| > \delta \right) \leq \frac{kE_0 D^{m/2}}{\delta^m}$$

$\square$

# B    PROOFS OF THEOREMS

*Proof of Theorem 2.* We calculate

$$
\begin{aligned}
\hat{\boldsymbol{\theta}} - \boldsymbol{\theta} &= (\mathbf{X}^\top \mathbf{X} + h_n \mathbf{I}_p)^{-1} \mathbf{X}^\top \boldsymbol{y} - \boldsymbol{\theta} \\
&= \mathbf{Q}(\boldsymbol{\Lambda}^2 + h_n \mathbf{I}_r)^{-1} \boldsymbol{\Lambda} \mathbf{P}^\top (\mathbf{P}\boldsymbol{\Lambda}\mathbf{Q}^\top \boldsymbol{\beta} + \boldsymbol{\epsilon}) - \mathbf{Q}\mathbf{Q}^\top \boldsymbol{\beta} \\
&= -h_n \mathbf{Q}(\boldsymbol{\Lambda}^2 + h_n \mathbf{I}_r)^{-1} \boldsymbol{\zeta} + \mathbf{Q}(\boldsymbol{\Lambda}^2 + h_n \mathbf{I}_r)^{-1} \boldsymbol{\Lambda} \mathbf{P}^\top \boldsymbol{\epsilon},
\end{aligned}
\tag{12}
$$

where $\boldsymbol{\zeta} = \mathbf{Q}^\top \boldsymbol{\beta}$. Denote $\widehat{\mathcal{M}}_{a_n} = \left\{ i \mid |\hat{\theta}_i| > a_n \right\}$, we have

$$
\mathrm{P}(\widehat{\mathcal{M}}_{a_n} \neq \mathcal{M}_{a_n}) \leq \mathrm{P}(\min_{i \in \mathcal{M}_{a_n}} |\hat{\theta}_i| \leq a_n) + \mathrm{P}(\max_{i \notin \mathcal{M}_{a_n}} |\hat{\theta}_i| > a_n)
$$

$$
\leq \mathrm{P}\left( \min_{i \in \mathcal{M}_{a_n}} |\hat{\theta}_i| - \max_{i \in \mathcal{M}_{a_n}} |h_n \sum_{j=1}^{r} \frac{q_{ij}\zeta_j}{\lambda_j^2 + h_n}| - \max_{i \in \mathcal{M}_{a_n}} |\sum_{j=1}^{r} \frac{q_{ij}\lambda_j}{\lambda_j^2 + h_n} \sum_{l=1}^{n} p_{lj}\epsilon_l| \leq a_n \right)
$$

$$
+ \mathrm{P}\left( \max_{i \notin \mathcal{M}_{a_n}} |\hat{\theta}_i| + \max_{i \notin \mathcal{M}_{a_n}} |h_n \sum_{j=1}^{r} \frac{q_{ij}\zeta_j}{\lambda_j^2 + h_n}| + \max_{i \notin \mathcal{M}_{a_n}} |\sum_{j=1}^{r} \frac{q_{ij}\lambda_j}{\lambda_j^2 + h_n} \sum_{l=1}^{n} p_{lj}\epsilon_l| > a_n \right),
\tag{13}
$$

From Cauchy inequality,

$$
\max_{i=1,\ldots,p} |h_n \sum_{j=1}^{r} \frac{q_{ij}\zeta_j}{\lambda_j^2 + h_n}| \leq \max_{i=1,\ldots,p} h_n \sqrt{\sum_{j=1}^{r} q_{ij}^2} \sqrt{\sum_{j=1}^{r} \frac{\zeta_j^2}{(\lambda_j^2 + h_n)^2}} = O(n^{\alpha_\theta - \delta}),
\tag{14}
$$

$$
\max_{i=1,\ldots,p} \sum_{l=1}^{n} (\sum_{j=1}^{r} \frac{q_{ij}\lambda_j}{\lambda_j^2 + h_n} p_{lj})^2 = \max_{i=1,\ldots,p} \sum_{j=1}^{r} \frac{q_{ij}^2 \lambda_j^2}{(\lambda_j^2 + h_n)^2} \leq \max_{i=1,\ldots,p} \frac{\sum_{j=1}^{r} q_{ij}^2}{\lambda_r^2}.
$$

Therefore, for sufficiently large $n$, from Assumption B5 and B6, we have

$$\min_{i \in \mathcal{M}_{a_n}} |\hat{\theta}_i| - \max_{i \in \mathcal{M}_{a_n}} |h_n \sum_{j=1}^{r} \frac{q_{ij}\zeta_j}{\lambda_j^2 + h_n}| - a_n \geq \frac{1}{2}(\frac{1}{c_a} - 1)a_n,$$

$$a_n - \max_{i \notin \mathcal{M}_{a_n}} |\hat{\theta}_i| - \max_{i \notin \mathcal{M}_{a_n}} |h_n \sum_{j=1}^{r} \frac{q_{ij}\zeta_j}{\lambda_j^2 + h_n}| < \frac{1}{2}(1 - c_a)a_n. \tag{15}$$

From lemma 4, there exist constants $E_1$ and $E_2$ depending on $m$ such that

$$\mathrm{P}(\widehat{\mathcal{M}}_{a_n} \neq \mathcal{M}_{a_n}) \leq \frac{|\mathcal{M}_{a_n}| \times E_1}{\lambda_r^m \times (\frac{1}{2}(\frac{1}{c_a} - 1)a_n)^m} + \frac{(p - |\mathcal{M}_{a_n}|) \times E_2}{\lambda_r^m \times (\frac{1}{2}(1 - c_a)a_n)^m} = O(n^{\alpha_p - m\eta + m\alpha_a}). \tag{16}$$

From Assumption B6, we verified the consistency of variable selection.

If $\widehat{\mathcal{M}}_{a_n} = \mathcal{M}_{a_n}$,

$$|X_i\hat{\boldsymbol{\theta}} - X_i\boldsymbol{\theta}| \leq |h_n \sum_{j=1}^{r} c_{ij} \frac{1}{\lambda_j^2 + h_n} \zeta_j| + |\sum_{j=1}^{r} c_{ij} \frac{\lambda_j}{\lambda_j^2 + h_n} \sum_{l=1}^{n} p_{lj}\epsilon_l| \tag{17}$$

From Cauchy inequality and lemma 4,

$$|h_n \sum_{j=1}^{r} c_{ij} \frac{1}{\lambda_j^2 + h_n} \zeta_j| \leq h_n \sqrt{\sum_{j=1}^{r} c_{ij}^2} \sqrt{\sum_{j=1}^{r} (\frac{1}{\lambda_j^2 + h_n} \zeta_j)^2} = O(n^{\alpha_\theta - \delta}). \tag{18}$$

$$\sum_{l=1}^{n} (\sum_{j=1}^{r} c_{ij} \frac{\lambda_j}{\lambda_j^2 + h_n} \sum_{l=1}^{n} p_{lj})^2 = \sum_{j=1}^{r} c_{ij}^2 \frac{\lambda_j^2}{(\lambda_j^2 + h_n)^2} \leq \frac{C_\mathcal{N}}{\lambda_r^2}$$

$$\Rightarrow P(|\sum_{j=1}^{r} c_{ij} \frac{\lambda_j}{\lambda_j^2 + h_n} \sum_{l=1}^{n} p_{lj}\epsilon_l| > \delta) \leq \frac{E(|\sum_{j=1}^{r} c_{ij} \frac{\lambda_j}{\lambda_j^2 + h_n} \sum_{l=1}^{n} p_{lj}\epsilon_l|)^m}{\delta^m} \tag{19}$$

$$\leq \frac{2^m C(m)E|\epsilon_1|^m (\sum_{j=1}^{r} c_{ij} \frac{\lambda_j}{\lambda_j^2 + h_n} \sum_{l=1}^{n} p_{lj})^{m/2})}{\delta^m} \leq \frac{2^m C(m)E|\epsilon_1|^m C_\mathcal{N}^{m/2}}{\delta^m \lambda_r^m}.$$

Choose a constant $C = 2^m C(m)E|\epsilon_1|^m$, we have $|\sum_{j=1}^{r} c_{ij} \frac{\lambda_j^2}{\lambda_j^2 + h_n} \sum_{l=1}^{n} p_{lj}\epsilon_l| = O_p(n^{-\eta})$. Therefore, we prove $|X_i\hat{\boldsymbol{\theta}} - X_i\boldsymbol{\theta}| = O_p(n^{\alpha_\theta - \delta} + n^{-\eta}) = O_p(n^{\alpha_\theta - \delta})$ according to Assumption B5.

Since $|Y_i - X_i\boldsymbol{\theta}| - |Y_i - X_i\hat{\boldsymbol{\theta}}| \leq |X_i\hat{\boldsymbol{\theta}} - X_i\boldsymbol{\theta}|$, for $\forall \epsilon \in (0, 1)$, there exist a constant $c_3 > 0$ such that

$$|Y_i - X_i\hat{\boldsymbol{\theta}}| - |Y_i - X_i\boldsymbol{\theta}| \leq c_3 n^{\alpha_\theta - \delta}, \quad i = 1, ..., n, \tag{20}$$

with at least $1 - \epsilon$ probability.

Denote $F_n$ as empirical CDF of $|Y_i - X_i\boldsymbol{\theta}|$, and $F_0$ is its distribution function. $F_{\hat{n}}$ is empirical CDF of $|Y_i - X_i\hat{\boldsymbol{\theta}}|$ and $F_1$ is its distribution function. In the following proof, we will achieve our conclusion through three main steps: first we clarify the relationship between $F_0^{-1}$ and $F_1^{-1}$. Next use DKW Theorem bound the discrepancy between the inverse of the empirical distribution and the true distribution, then analyze the relationship between the two empirical distributions, and finally combine the results of the previous steps to conclude the proof.

From (20) we have

$$|F_1(t) - F_0(t)| = P(|Y_i - X_i\hat{\boldsymbol{\theta}}| < t) - P(|Y_i - X_i\boldsymbol{\theta}| < t)$$
$$\leq P(|Y_i - X_i\boldsymbol{\theta}| - c_3 n^{\alpha_\theta - \delta} < t) - P(|Y_i - X_i\boldsymbol{\theta}| < t) \tag{21}$$
$$\leq r c_3 n^{\alpha_\theta - \delta}.$$

On the event $|Y_i - X_i\hat{\boldsymbol{\theta}}| - |Y_i - X_i\boldsymbol{\theta}| \leq c_3 n^{\alpha_\theta - \delta}$, using lemma 1 and lemma 2 we obtain $|F_1^{-1}(1 - \alpha) - F_0^{-1}(1 - \alpha)| \leq c_4 n^{\alpha_\theta - \delta}$, where $c_4 > 0$ is a constant. According to lemma 3, we obtain

$$|F_{\hat{n}}^{-1}(1 - \alpha) - F_n^{-1}(1 - \alpha)| = O_p(n^{\alpha_\theta - \delta}). \tag{22}$$

Applying the Dvoretzky–Kiefer–Wolfowitz (DKW) inequality, we have

$$\begin{aligned} |F_n^{-1}(1 - \alpha) - F_0^{-1}(1 - \alpha)| &= O_p(n^{-1/2}), \\ |F_{\hat{n}}^{-1}(1 - \alpha) - F_1^{-1}(1 - \alpha)| &= O_p(n^{-1/2}). \end{aligned} \tag{23}$$

Therefore, combining with above inequalities (21), (22), (23) and (16), we have $|F_{\hat{n}}^{-1}(1 - \alpha) - q_{1-\alpha}| = O_p(n^{\alpha_\theta - \delta})$. $\square$

*Proof of Theorem 3.* First we define

$$\tilde{\boldsymbol{\theta}} = \hat{\boldsymbol{\theta}}^* + h_n X_i \mathbf{Q}(\boldsymbol{\Lambda}^2 + h_n \mathbf{I}_r)^{-1} \mathbf{Q}^\top \hat{\boldsymbol{\theta}}^*, \tag{24}$$

and the debiased conformal prediction band is given by

$$X_n \tilde{\boldsymbol{\theta}} \pm \text{ the } \lceil (1 - \alpha)(n + 1) \rceil \text{-th smallest of } (\tilde{R}_1^y, \ldots, \tilde{R}_{n+1}^y).$$

We calculate

$$\tilde{\boldsymbol{\theta}} - \boldsymbol{\theta} = -h_n^2 \mathbf{Q}(\boldsymbol{\Lambda}^2 + h_n \mathbf{I}_r)^{-2}\boldsymbol{\zeta} + \mathbf{Q}\big((\boldsymbol{\Lambda}^2 + h_n \mathbf{I}_r)^{-1}\boldsymbol{\Lambda} + h_n(\boldsymbol{\Lambda}^2 + h_n \mathbf{I}_r)^{-2}\boldsymbol{\Lambda}\big)\mathbf{P}^\top \boldsymbol{\epsilon}. \tag{25}$$

Following the analysis of Theorem 2, we have

$$|X_i \tilde{\boldsymbol{\theta}} - X_i \boldsymbol{\theta}| = O_p(n^{\alpha_\theta - 2\delta} + n^{-\eta}) = O_p(n^{-\eta}), \tag{26}$$

where $\alpha_\theta - 2\delta < -\eta$ according to Assumption B5, and $\alpha_\theta$ and $\delta$ are constants defined under the assumptions.

Since $|Y_i - X_i\boldsymbol{\theta}| - |Y_i - X_i\tilde{\boldsymbol{\theta}}| \leq |X_i\tilde{\boldsymbol{\theta}} - X_i\boldsymbol{\theta}|$, denote $F_{\tilde{n}}$ as the empirical CDF of $|Y_i - X_i\tilde{\boldsymbol{\theta}}|$, and $F_3$ as its corresponding distribution function. For any $\epsilon \in (0, 1)$, there exists a constant $c_5 > 0$ such that

$$||Y_i - X_i\tilde{\boldsymbol{\theta}}| - |Y_i - X_i\boldsymbol{\theta}|| \leq c_5 n^{-\eta}, \quad i = 1, \ldots, n, \tag{27}$$

with at least $1 - \epsilon$ probability.

Similarly to Theorem 2, we derive

$$\begin{aligned} |F_3(t) - F_0(t)| &= P(|Y_i - X_i\tilde{\boldsymbol{\theta}}| < t) - P(|Y_i - X_i\boldsymbol{\theta}| < t) \\ &\leq P(|Y_i - X_i\boldsymbol{\theta}| - c_5 n^{-\eta} < t) - P(|Y_i - X_i\boldsymbol{\theta}| < t) \\ &\leq r c_5 n^{-\eta}, \end{aligned} \tag{28}$$

where $r$ is a constant defined under the assumptions.

Using Lemma 1 and Lemma 2, we establish

$$|F_3^{-1}(1 - \alpha) - F_0^{-1}(1 - \alpha)| = O_p(n^{\alpha_\theta - 2\delta}). \tag{29}$$

Applying Lemma 3, we deduce

$$|F_{\tilde{n}}^{-1}(1 - \alpha) - F_n^{-1}(1 - \alpha)| = O_p(n^{\alpha_\theta - 2\delta}). \tag{30}$$

Finally, invoking the Dvoretzky–Kiefer–Wolfowitz (DKW) inequality, we have

$$|F_{\hat{n}}^{-1}(1 - \alpha) - F_3^{-1}(1 - \alpha)| = O_p(n^{-1/2}). \tag{31}$$

Combining these results, we conclude that

$$\text{Width}(\hat{C}_{\text{ThCRR}}^{\text{Debias}}(X)) - 2q_{1-\alpha} = O_p(n^{-\eta}).$$

$\square$

*Proof of Theorem 4.* We follow the methodology outlined in Theorem 3.4 of Lei et al. (2018). The proof is divided into two main parts: first, we demonstrate that the center of the prediction interval derived from our method is asymptotically close to the center of the oracle prediction interval; second, we show that the lengths of the two intervals are also asymptotically equivalent.

Indeed, we establish that the center of the prediction interval, $X_{n+1}\tilde{\theta}$, is close to the oracle center, $X_{n+1}\theta$. This claim can be rigorously verified using Equation (26). Next, we analyze the length of the prediction interval. This part directly follows from Theorem 3, which provides the necessary bounds and asymptotic equivalence for interval lengths.

By combining these two results, we conclude that the prediction interval constructed by our method asymptotically matches the oracle prediction interval in both center and length, completing the proof.

$\square$

## C  ADDITIONAL EXPERIMENTAL RESULTS

We present some additional experiments here. As mentioned in Section 4, we conduct conformalized threshold ridge regression, debiased conformalized threshold ridge regression, and debiased threshold ridge regression using bootstrap on seven benchmark datasets. Details about mean and standard deviation are presented in Table 1, visual results about *bike*, *concrete* and *facebook_2* can be seen in Fig 4. These datasets are described in Section 4.

To demonstrate performance of our method, we measure conditional coverage on three datasets, *bike*, *community* and *concrete*, which include the cases $n < p$ and $n > p$. And we compare our method with more different methods: conformalized threshold ridge regression, debiased conformalized threshold ridge regression, boostrap, conformalized quantile regression, NuSVR and split conformalized ridge regression. Results are shown in Fig 7, 8 and 9.

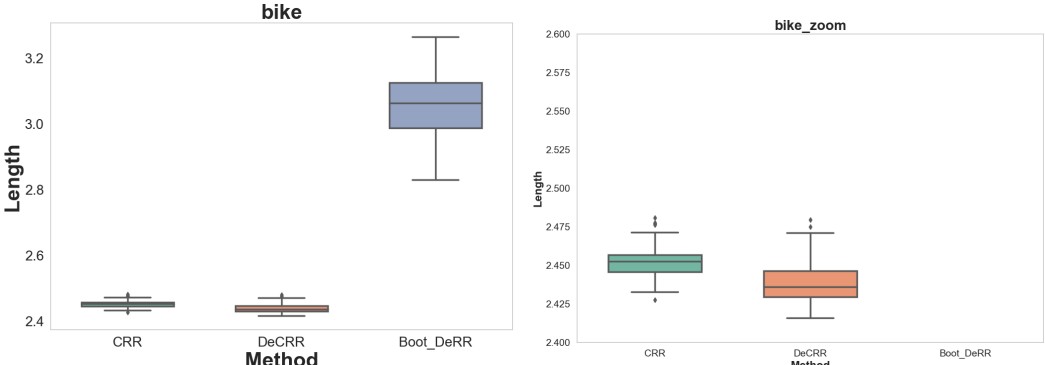

Figure 4: Length of prediction intervals on the benchmark dataset *bike*.

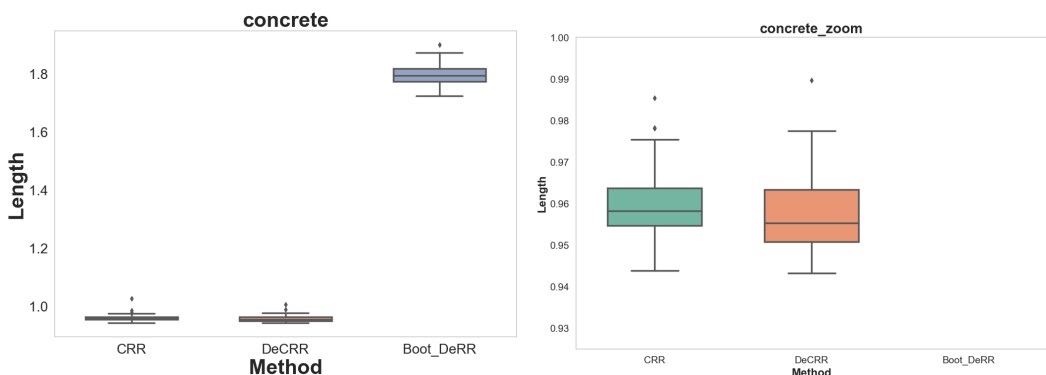

Figure 5: Length of prediction intervals on the benchmark dataset *concrete*.

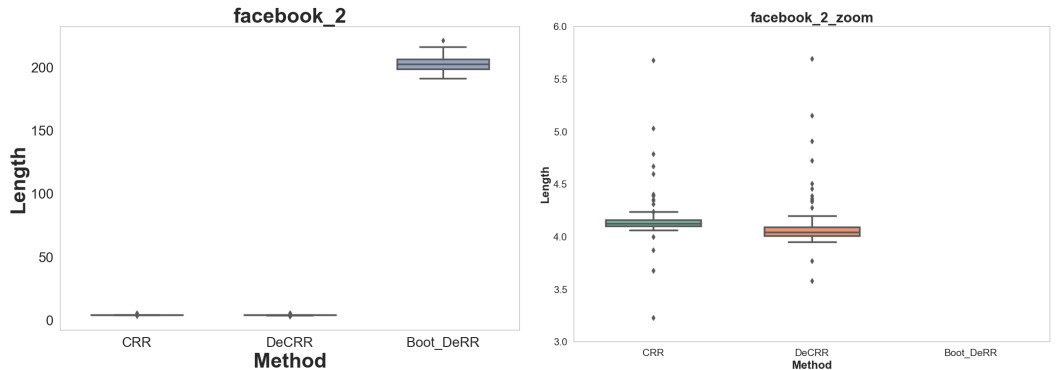

Figure 6: Length of prediction intervals on the benchmark dataset *facebook_2*.

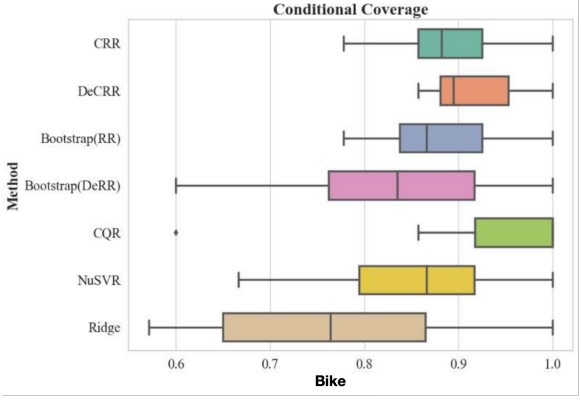

Figure 7: Conditional coverage of prediction intervals on the benchmark dataset *bike*.

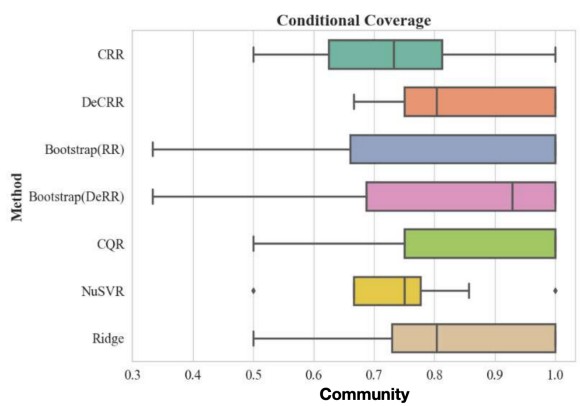

Figure 8: Conditional coverage of prediction intervals on the benchmark dataset *concrete*.

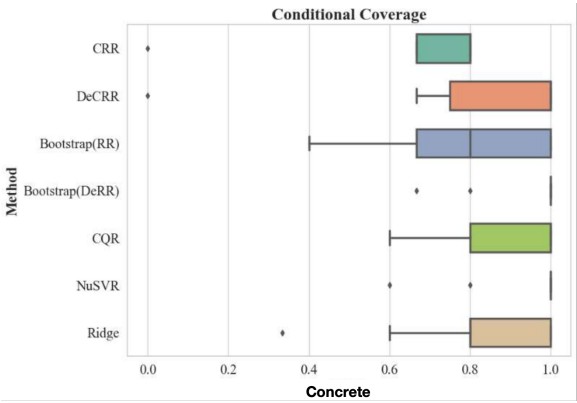

Figure 9: Conditional coverage of prediction intervals on the benchmark dataset *facebook_2*.

