# OpenReview forum: "Conformal Prediction with Model-Aware Debiasing"
_ICLR.cc/2025/Conference — ICLR 2025 Conference Withdrawn Submission_

### Official Review · Reviewer_V4xL · 2024-11-02

**Soundness:** 2
**Presentation:** 2
**Contribution:** 2
**Rating:** 3
**Confidence:** 4

**Summary:**

This paper studies the efficiency of conformal prediction sets. In particular, the paper argues that the conformal prediction intervals
can be wide due to the bias of point predictions. It is then proposed that by debiasing the point prediction, one can reduce the length of the conformal prediction interval. The discussion is under the linear model (this is inferred from Assumption B.3; please correct me if I am mistaken) with the ridge regression prediction model. The proposed method is evaluated and compared with other methods in numerical studies.

**Strengths:**

This work considers an important question (efficiency of predictive inference), and has made some interesting observations that could potentially lead to improvement.

**Weaknesses:**

Weaknesses:
1.  The proposed method is restrictive in the following sense: (1) the discussion is under the linear model (this is inferred from Assumption B.3; please correct me if I am wrong); (2) the prediction method is ridge regression; (3) there are several assumptions needed for the provable efficiency improvement. Under these conditions, there can be better ways to construct prediction intervals (which has not been discussed in the current manuscript).

2. The numerical results are not convincing for the paper's argument. First, in several settings, the improvement over CRR is not significant (after accounting for the variability). Second, there are many other ways to construct nonconformity scores (e.g., conformalized quantile regression [1], conformalized distributional regression [2]) that have not been compared in the experiments.

[1] Romano, Yaniv, Evan Patterson, and Emmanuel Candes. "Conformalized quantile regression." Advances in neural information processing systems 32 (2019).
[2] Chernozhukov, Victor, Kaspar Wüthrich, and Yinchu Zhu. "Distributional conformal prediction." Proceedings of the National Academy of Sciences 118.48 (2021): e2107794118.

**Questions:**

My main questions are listed in the weaknesses section. There are
a few minor questions:

1. In A2, is it supposed to be that "the density of $\epsilon$ is symmetric about 0 and nonincreasing on $\mathbb{R}_+$"? (Otherwise, the density function is a constant function.)
2. It is stated that "Assumption A1 ensures that the data satisfies the condition of exchangeability without imposing the requirement of independence". But A1 only requires the marginal distribution of $(X,Y)$ to be the same, which does not imply
exchangeability. For example, $(X_1,Y_1) =  (X_2,Y_2)$ and $(X_3,Y_3)$ independent of
$(X_1,Y_1,X_2,Y_2)$.
3. On line 193, there are repeated referral to (3).
4. It is stated in line 199 that "Recall the conformal prediction band constructed in Section 2.2, the width of band is $2T_{1−\alpha}(|Y_i −\hat Y_i|) ...$". This is not true for, e.g., the standardized absolute fitted residual.

---

### Official Review · Reviewer_U8d2 · 2024-11-03

**Soundness:** 3
**Presentation:** 3
**Contribution:** 3
**Rating:** 6
**Confidence:** 4

**Summary:**

The paper introduces a "model-aware conformal prediction" method that aims to solve the problem of wide prediction intervals caused by model bias. Unlike existing approaches that tackle bias, usually requiring strict assumptions, their method only requires exchangeability, guarantees finite-sample coverage, and produces tighter intervals.
They explore the methods properties via a theoretical analysis, and a simulation study.

**Strengths:**

The contribution is indeed fairly novel. To my knowledge, the authors are the first in tackling explicitly the issue of model bias in conformal prediction, and they tackle it in a fairly structured and interesting way.

**Weaknesses:**

My main concern about the paper lies in the simulation. The proposal is surely interesting, yet a reader is left wanting way more. The authors focus themselves only on a limited number of real world datasets. This is even more true taking into account the fact that in some of the proposed tests the impact of debiasing (I would say the main contribution of this work) seems to provide very limited benefits with respect to non-debiased alternatives.

**Questions:**

Can you provide a better "simulative" case for the competitiveness of your method? How does the method perform on other datasets? Can you identify specific "stylised" situations (by data simulation) where your model clearly shows to be more competitive than the alternatives?

---

### Official Review · Reviewer_udF2 · 2024-11-03

**Soundness:** 3
**Presentation:** 1
**Contribution:** 2
**Rating:** 3
**Confidence:** 5

**Summary:**

The paper addresses the limitations of current prediction methods, which often result in wide, inefficient prediction intervals due to biases in model estimation. These biases can lead to conservative predictions that reduce the usefulness of the intervals in practical applications. The authors propose a novel model-aware conformal prediction method that integrates model information to mitigate biases without relying on strict assumptions about the data distribution. This method promises more accurate prediction intervals with reliable finite-sample coverage in various regression applications.

The main contributions are
- The authors present a model-based debiasing approach in the context of conformal prediction. By directly accounting for biases in the nonconformity score function, shorter prediction intervals are generated and thus prediction efficiency is improved while finite-sample coverage is preserved.
- To demonstrate the practical value of their method, the authors apply it to threshold ridge regression, a situation where computational simplicity is an advantage. They prove that their model-dependent approach converges to the oracle prediction band under certain conditions and achieves asymptotic conditional validity.
- Theoretical and empirical evaluation: The paper provides a theoretical validation for the model-based conformal prediction method, focusing on finite-sample marginal coverage and asymptotic conditional validity. In addition, the performance of the method was compared with other approaches, showing efficiency gains due to shorter prediction intervals across multiple data sets.

The work mainly builds on Zhang & Politis (2022), who used bias correction in threshold ridge regression to improve prediction interval performance, relying on a hybrid bootstrap with asymptotic coverage guarantees. Zhang and Politis approach is asymptotic it may not be robust enough for finite samples. The proposed method extends their debiasing technique within a conformal prediction framework and guarantees marginal coverage for finite samples without stringent assumptions, which improves practical reliability. Moreover, the authors prove that the prediction intervals converge to the oracle prediction band under certain conditions and achieve asymptotic conditional validity.

The authors tested the model-aware conformal prediction on several real-world datasets in scenarios where the number of features exceeded the number of observations (high-dimensional setting) and where the sample size exceeded the number of features.

**Strengths:**

- This article deals with the debiasing of predictors, a topic that has attracted  interest, especially in areas where sparse regression methods are widely used.
- The paper aims to provide theoretical results on debiasing methods, which is an ambitious and relevant endeavor.*
-

**Weaknesses:**

The paper makes ambitious promises, but fails to make any meaningful contributions. While the notion of "debiasing" a predictor is intuitively reasonable and a fruitful area of research, especially in sparse regression, the results in this paper are surprisingly unconvincing. Theorem 1 offers absolutely no new insights, lacking substance and applicability. Moreover, Theorem 2 is marred by an overly complex framework that relies on no less than ten assumptions. Each of these assumptions is fundamentally unverifiable in practice, and in most real-world applications they are systematically violated. This makes it extremely difficult to recognize the practical or theoretical relevance of Theorem 2.

The presentation of the paper also raises concerns. It was obviously produced in haste and suffers from poor proofreading, with numerous instances of awkwardly worded sentences that detract from clarity. For example:
 - There is an unaddressed placeholder "according to xxx" in the supplemenent— an oversight that undermines the professionalism of the paper.
- The authors claim that assumption  A1  "ensures that the data fulfill the condition of interchangeability"," which is absolutely false as written
- Theoprem 3 is not proven "the rest  of the proof is the same than Theorem 2" which is mysterious, given that the conclusion differs. Given the assumption, I think it is possible to give an expression for $\eta$.

These problems with language and clarity run throughout the text. I could cite numerous other examples of clumsy wording and convoluted explanations. And while I suspect there are technical errors as wel, making it extremely difficult to follow the logical flow. In ** Lemma 2 **, for example, the author seems to claim that (F_1 (F_1-1 (q)) = q) without any assumptions about (F_1), leaving important conditions unconsidered. This lack of rigor and precision is characteristic of the entire paper and makes it difficult to extract coherent insights or valid contributions from the results presented.

**Questions:**

- Polish the proof and provide an proof of Theorem 3 with an explicit constant \eta
- In Eq. (25), it is said that \left|X_i \hat{\theta}-X_i \theta\right|=O_p\left(n^{\alpha_\theta-\delta}+n^{-\eta}\right)=O_p\left(n^{-\eta}\right), whereas it said in the main text that "Theorem 2 demonstrates that the conformal interval converges to the oracle prediction band if $\delta$ as $n \rightarrow \infty$. The proof is presented in Appendix B. Unfortunately, $\alpha_\theta$ is typically not less than $\delta$... There is something wrong. I would like to see a proof of heorem 4

---

### Official Review · Reviewer_9sf3 · 2024-11-04

**Soundness:** 2
**Presentation:** 2
**Contribution:** 2
**Rating:** 3
**Confidence:** 3

**Summary:**

The authors propose a new conformal prediction method for threshold ridge regression. It is based on a new nonconformity score function that uses the estimate of the model's bias. This approach is claimed to provide shorter valid prediction intervals than those by classic conformal prediction and also to be asymptotically conditionally valid. Authors perform numerical experiments on public datasets to illustrate their findings.

**Strengths:**

- Authors propose a new technique based on conformal prediction that addresses some shortcomings of the basic conformal prediction method that were argued by prior work.
- The results are adequately explained.

**Weaknesses:**

- I think that Theorems 2-4 might be not correct. In particular, Theorem 2 uses Holder continuity with respect to inverse of empirical CDF. Empirical CDF is not even continuous, so the usage of Holder continuity looks not correct.

- Authors claim in Conclusion that their new method provides “stronger validity, including asymptotic conditional coverage”. I think the conditional coverage is not properly checked with experiments. Even if the required conditions are not fully satisfied in practice, it is still valuable to see to which extent the conditional coverage is achieved.

- Limited number of baseline methods to compare against. The proposed approach requires cross-validation to tune the parameters, which increases the computational cost. This brings this method closer in resource demands to other methods like neural network quantile regression, etc.

- Figure 3 is really hard to look at, the numbers and marks are too small. Consider improving the visual presentation of the experimental results.

- It appears that the notation in sections 2-3  is a mix of full and split conformal. The whole section 2.2 can be improved for more clarity.
- In formula (2), what is $\mu^{y}$?
- In formula (5), it should be until (n). Also, dependence on $y$ seems strange.
- Section 3, again the same mistake. It should be either ceil((1-\alpha)(n+1))-th value of the $n$ sorted calibration scores or level $(1-\alpha)$ quantile of (the distribution defined by ) the sum of (n+1) deltas.


#### Comment after rebuttal #####

Theory was partially corrected by authors. How it still contains multiple misprints, wrong references, ...

**Questions:**

- You comment on assumption A1 that it "ensures that the data satisfies the condition of exchangeability without imposing the
requirement of independence". However, if $(X_i, Y_i)$ are not i.i.d., what Assumption A1 states then? It doesn't describe any particular data generation algorithm
- How do you apply Holder continuity to empirical CDF?
- Have you checked the conditional validity experimentally? Is it achieved for any of the considered dataset?
- The title of the paper does not mention ridge regression specifically. How can  “model-aware de-biasing” be applied to a wider class of prediction models?

---

### Note · Authors · 2025-03-24

I have read and agree with the venue's withdrawal policy on behalf of myself and my co-authors.

---

### Meta-Review · Area_Chair_9QKS · 2024-12-21

**Metareview:**

The core problem addressed in the paper is the issue of bias in model estimation (e.g. due to errors in the estimated model parameters), which can lead to overly wide or inefficient prediction intervals in predictive inference, reducing their utility.

The core insight is to utilize **known** (by design) model information (e.g., structure on the desired parameters) for debiasing and **unknown** aspects (e.g., data distribution) to be addressed by the conformal prediction framework. This formally achieved by introducing a Bias-Corrected Nonconformity Score Construction. Namely instead of the classical score in regression problem: $r_i = |y_i - \hat \mu(x_i)|$, introduce a bias correction term removing the systematic error due to model bias i.e.  $\tilde{r}_i = |y_i - \hat \mu(x_i) + b(x_i)|$.
Bias term is in general defined as $b(x) = \mathbb{E}[\hat \mu(x)] - \mu(x)$ which is controlled under classical linear regression assumption and when using Ridge regression as base model. The overall validity is straightforward as long as the bias term respect the excheangeability of the scores.

While acknowledging that the core idea is exciting, the reviewers were not convinced by the delivery mentioning explicit technical flaws in the paper along with convoluted presentations. After reading the discussions (see summary of main point raised below) and papers, I agree with the reviewers and recommend a *reject*.

I encourage the authors to pursue this line of research, which I consider very interesting and also rewrite the paper in a more transparent and clear term. Many points are very easy to describe but the authors phrasing is more than hard to read. Finally, I also recommend to tone down some of the claims. Controlling the bias term is very hard in practice and depends on tones of assumptions e.g. how would this be done for neural network base model?

**Additional Comments On Reviewer Discussion:**

- *Reviewer 9sf3*: Criticized the use of Hölder continuity with respect to the inverse empirical CDF in Theorem 2, stating it is inappropriate as empirical CDFs are not continuous. Found multiple issues in theoretical derivations and requested corrections, lacks sufficient proof, referencing incorrect assumptions. The claim of achieving asymptotic conditional coverage was questioned due to insufficient experimental validation. Highlighted unclear notations, missing definitions, and incorrect references in formulas. Found some Figure hard to interpret due to poor visual design. Suggested the inclusion of more baseline methods for comparison and pointed out the specific computational overhead.

- *Reviewer udF2*: Over-Promising Results and critiqued the practicality of the results, especially mains Theorems rely on no less than ten unverifiable assumptions in real-world applications, plus lack novelty along with overly complex presentation. Highlighted poor proofreading, including placeholder text and unclear explanations. Questioned the rigor of proofs.
Overall limited applicability of the method due to restrictive assumptions and a focus on ridge regression. It is also important to add that the reviewer acknowledged the paper merit but needs careful reconsideration.

- *Reviewer U8d2*: Found the empirical evaluation insufficient, with few datasets tested and limited scenarios to demonstrate the method’s benefits. Observed that the debiasing approach provided minimal improvements in some tests, questioning its practical significance.
Suggest to Include more datasets and stylized simulations to highlight the method’s competitive edge.

- *Reviewer V4xL*:
Criticized the narrow focus on linear models and ridge regression, limiting the method’s broader applicability.
Noted that assumptions for provable efficiency are restrictive and questioned their practical validity.
Found the numerical results unconvincing due to minor improvements over baseline methods like CRR.
Mentioned that alternative methods for constructing nonconformity scores (e.g., conformalized quantile regression) were not included for comparison. Suggested to address applicability to non-linear and non-parametric settings.
Compare the method with alternative approaches beyond CRR.


Overall, despite these critical negative points, all reviewers acknowledged the importance of the research question (efficiency of predictive inference) and the potential for improvement. I align with them, the current version of the paper is not ready for publication.

---

### Decision · Program_Chairs · 2025-01-22

Reject